# High-Spatial Resolution Maps of PM2.5 Using Mobile Sensors on Buses: A Case Study of Teltow City, Germany, in the Suburb of Berlin, 2023

Jean-Baptiste Renard [1,*], Günter Becker [2], Marc Nodorft [3], Ehsan Tavakoli [3], Leroy Thiele [4], Eric Poincelet [5], Markus Scholz [3] and Jérémy Surcin [5]

1   LPC2E-CNRS, 3A Avenue de la Recherche Scientifique, Cedex 2, 45071 Orléans, France
2   RISA Sicherheitsanalysen GmbH, 10707 Berlin, Germany; guenter.becker@risa.de
3   DEUS Pollutrack Smart City GmbH IG, 14513 Teltow, Germany; m.nodorft@deus-pollutrack.com (M.N.); e.tavakoli@deus-cleangrid.org (E.T.); scholz@novelsense.com (M.S.)
4   Industrieelektronik Brandenburg GmbH, 14772 Brandenburg, Germany; l.thiele@ieb.gmbh
5   Pollutrack, 5 rue Lespagnol, 75020 Paris, France; e.poincelet@gmail.com (E.P.); jeremysurcin.pollutrack@protonmail.com (J.S.)
*   Correspondence: jean-baptiste.renard@cnrs-orleans.fr

**Abstract:** Air quality monitoring networks regulated by law provide accurate but sparse measurements of PM2.5 mass concentrations. High-spatial resolution maps of the PM2.5 mass concentration values are necessary to better estimate the citizen exposure to outdoor air pollution and the sanitary consequences. To address this, a field campaign was conducted in Teltow, a midsize city southwest of Berlin, Germany, for the 2021–2023 period. A network of optical sensors deployed by Pollutrack included fixed monitoring stations as well as mobile sensors mounted on the roofs of buses and cars. This setup provides PM2.5 pollution maps with a spatial resolution down to 100 m on the main roads. The reliability of Pollutrack measurements was first established with comparison to measurements from the German Environment Agency (UBA) and modelling calculations based on high-resolution weather forecasts. Using these validated data, maps were generated for 2023, highlighting the mean PM2.5 mass concentrations and the number of days per year above the 15 µg.m$^{-3}$ value (the daily maximum recommended by the World Health Organization (WHO) in 2021). The findings indicate that PM2.5 levels in Teltow are generally in the good-to-moderate range. The higher values (hot spots) are detected mainly along the highways and motorways, where traffic speeds are higher compared to inner-city roads. Also, the PM2.5 mass concentrations are higher on the street than on the sidewalks. The results were further compared to those in the city of Paris, France, obtained using the same methodology. The observed parallels between the two datasets underscore the strong correlation between traffic density and PM2.5 concentrations. Finally, the study discusses the advantages of integrating such high-resolution sensor networks with modelling approaches to enhance the understanding of localized PM2.5 variability and to better evaluate public exposure to air pollution.

**Keywords:** PM2.5; mass concentrations; urban traffic; mobile sensors; Teltow city; Germany





## 1. Introduction

Measuring the time evolution of PM2.5 (particulate matter smaller than 2.5 µm) in ambient air is of high importance, since it has both short-term and long-term adverse effects on human health, from respiratory disorders and heart attacks to neurodegenerative diseases [1–5]. In particular, recent studies have shown the direct linear effect of PM2.5 urban pollution on COVID-19 mortality, with an increase in mortality up to 40% per 1 µg.m$^{-3}$ increase in PM2.5 mass concentrations [6,7].

The local content of PM pollution can be strongly variable due to the heterogeneity of sources, the presence of imported particles, and the formation of secondary aerosols [8–10].

These particles are mainly anthropogenic, although some natural sources, such as desert dust or biomass fire smoke events, can be considered. PM can be horizontally and vertically transported from a local to continental scale, and their concentration is strongly related to meteorological conditions. Thus, there is a need for accurate local PM measurements to better evaluate the exposure of populations to pollution during their outdoor activities and to reduce the "hot spots" of pollution.

The WHO (World Health Organization) released in September 2021 new recommendations for PM2.5 mass concentrations, where the mean annual value must remain below 5 $\mu g.m^{-3}$ and the daily values must not exceed 15 $\mu g.m^{-3}$ more than 4 days per year [11]. Based on health studies, such recommendations can be the baseline for analyzing PM2.5 pollution.

The law-regulated PM10 and PM2.5 measurements are conducted by dedicated organizations, such as the German Environment Agency (UBA) for Germany [12]. Because of the instrument and high operational costs, no more than a few automatic monitoring stations with an accuracy typically of a few $\mu g.m^{-3}$ [13] are available per city or region. With such a network, it is neither possible to study the PM spatial variability at the local scale nor identify the hot spots beyond the vicinity of the stations.

In the case of relatively flat landscapes, one can expect that these sparse permanent measurements or those from temporary fixed stations installed during a field campaign, as performed in the Berlin region (Germany) [14], can provide a good proxy of the mean PM mass concentration for a surface of hundreds of $km^2$. For dense cities like Paris (France) with a large number of canyon streets, hills, and surrounding motorways, strong local variabilities at a scale of a kilometre may occur [15,16]. Permanent or occasional urban pollution spots with different origins, such as fires, commercial food activities, traffic jams, and construction sites can also occur. To detect them, there is a need to conduct permanent measurements with a spatial resolution as accurate as possible, down to the width of a street.

Alternatively, modelling approaches could be used to estimate the local values of PM pollution. Different modelling works have given very promising results from a local scale, for instance, the Berlin region [17,18], up to the European continental scale [19]. However, such models need accurate registers of the permanent emission sources and local wind velocity and direction, and they cannot predict the contribution of non-permanent sources. More sophisticated modelling works are proposed, where the streets, the hourly traffic volume, and the mean speed of vehicles [20] are considered. Nevertheless, such approaches need to be validated for a large number of different conditions, using in situ measurements inside the traffic flow and on sidewalks.

For this purpose, medium-cost mobile optical sensors have been mounted since 2018 on the roof of some electric cars of the national electricity grid operator Enedis, then of the parcel delivery group Geopost/DPD. This system, called the Pollutrack network, is operated in more than 30 large European cities to provide high-spatial resolution maps of PM2.5 pollution. The need for daily recurrent measurements on the same streets or zones to statistically provide a resolution of 1 km or better implies using a large fleet of vehicles. The first results from the Pollutrack network were published for Paris, France, using up to 400 vehicles [15,16], showing the local effect of the motorways, as well as the consequences of the city's topography.

A combination of measurements from accurate reference stations, low- or medium-cost mobile sensors, and modelling tools could be the best approach to first cross-validate the consistency of the different sources of data and, secondly, to better evaluate the daily exposure of the citizens to local pollution. Towards this future strategy, we present here the results of PM2.5 mass concentrations obtained with the DEUS-SmartAir Network during a field campaign conducted around the midsize city of Teltow (~27,000 inhabitants in 2020) in the southwest of Berlin, Germany. The campaign is conducted with fixed and mobile sensors from the beginning of 2021 till the end of 2023. Compared to the measurements in the highly populated city of Paris, this new campaign should identify the presence of

hot spots in a moderately populated city. While the maps of PM2.5 for Paris have a spatial resolution of 1 km, we propose here to study pollution at the street scale. In the Teltow district, Pollutrack sensors are operated by the DEUS-Pollutrack team and are installed in their air quality monitoring device.

This analysis aims to build maps of the mean PM2.5 pollution levels and establish the number of days per year above the WHO recommendations. All the data (sensors, references, and modelling) will be first integrated daily for cross-comparison and then will be averaged to produce PM2.5 maps for the year of 2023.

## 2. Materials and Methods

### 2.1. Instrumentation

The UBA reference station for the PM2.5 measurements is located at Blankenfelde-Mahlow, 10 km west of the centre of Teltow (Table 1 and Figure 1). Data are available every 30 min. The data here are from 1 January 2021 to 31 December 2023. The federal office for environmental protection of Germany (Umweltbundesamt, UBA) relies on measuring stations with established technology validated for regulatory purposes. Data from these stations can therefore be used for the validation of other sensors.

**Table 1.** Locations of the fixed sensors.

| Station | Latitude (° N) | Longitude (° E) | Position |
|---|---|---|---|
| UBA reference | 52.349703 | 13.424306 | |
| Pollutrack 1 | 52.387226 | 13.302431 | House wall |
| Pollutrack 2 | 52.385992 | 13.301668 | House wall |
| Pollutrack 3 | 52.386310 | 13.301871 | Pole in a garden |
| Pollutrack 4 | 52.390000 | 13.302722 | House wall |

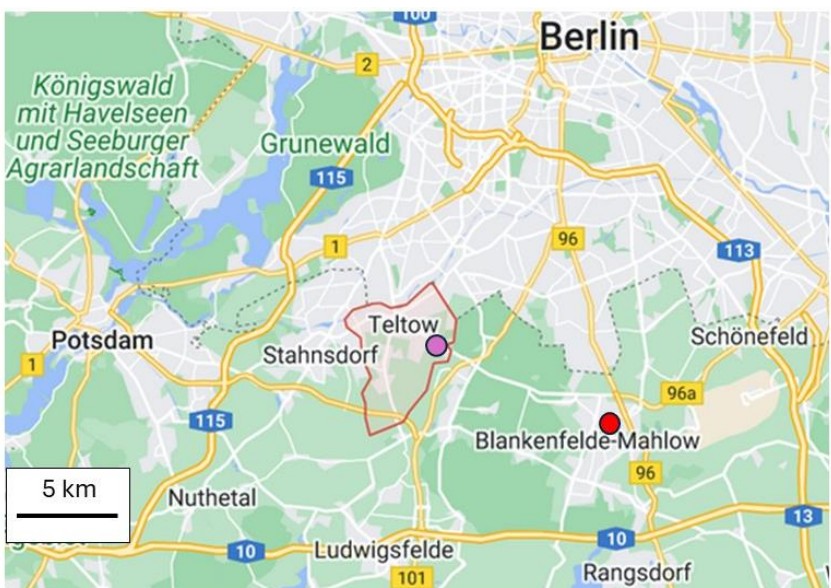

**Figure 1.** A map of the south suburb of Berlin, including Teltow; the red dot represents the PM2.5 reference station and the purple dot represents Pollutrack fixed stations (north is up).

A set of 4 Pollutrack sensors from the Pollutrack Company (France) were installed close together to the location "Heinersdorf" in the west–southwest of Teltow city (Table 1 and Figure 1), 2 km from the centre of the city and 8 km from the UBA reference station. The measurements cover the same period.

The Pollutrack sensors, developed and provided by the Pollutrack company (France), are optical aerosol counters that provide PM2.5 mass concentrations and number concentrations for 5 size classes in the 0.5–10 μm size range [15] every 30 s. The latest version of

the data is considered here, where the mass concentrations have been corrected to mitigate the humidity effect that decreases the mean density of the particles when they are hydrated. The data are processed onboard the sensors to provide number and mass concentrations. They are sent in real time by GSM to the Pollutrack data centre, where they are stored. The data are then reprocessed, taking into account humidity values obtained from the nearest weather station. Finally, the data are averaged over 30 min.

The sensors were also validated to perform measurements from the roof of cars driving in urban conditions, with speeds of up to several tens of km.h$^{-1}$. The shape of the instrument was specially designed, with the inlet at the opposite of the vehicle motion (Figure 2). Instead of using the vehicles of a parcel delivery company, as performed for other European cities with Geopost/DPD, 36 sensors were mounted here on the roof of city bus lines (Regiobus Potsdam Mittelmark GmbH), while 8 sensors were mounted on the roofs of municipality vehicles and of McDonald's cars. The data here are from 1 January 2023 to 31 December 2023.

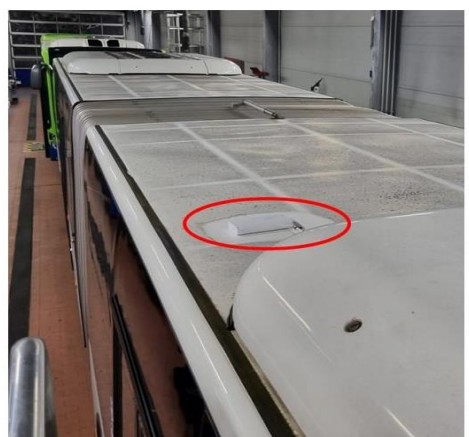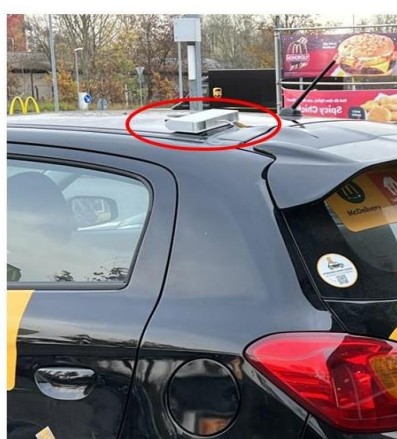

**Figure 2.** Pollutrack sensors (inside the red circle) on the roof of a bus (**left**) and of a car (**right**).

For such applications, time, GPS position, and Pollutrack measurements are available every 10 s. As for fixed stations, the data are sent by GSM to the Pollutrack data centre and are reprocessed using humidity data from the closest weather station. Finally, the data are hourly averaged for a square of 100 m in edge length.

### 2.2. Modelling

Modelling calculations on the PM2.5 concentrations were conducted for 2022. The calculations are based on the PM prediction model using high-resolution weather forecasts for the temperature profile, wind, and precipitation [21]. This approach implicitly assumes that meteorological parameters are constant over a few kilometres and can be sufficient for PM forecasting. It is suitable for Teltow, which is relatively flat, with altitudes varying between 35 m and 55 m. No strong local wind and temperature effects are present, contrary to what can happen over steep terrains. The model uses a neural network approach; the learning routine is based on a backpropagation algorithm. This approach was originally focused on PM10 measurements and was subsequently adapted to the study of PM2.5. It has been validated in [22] by comparing results with 11 models' outputs published between 2000 and 2021.

### 2.3. Cross-Comparisons

When using a large number of copies of the same instrument, it is necessary to evaluate their reproducibility. A previous study has shown that the discrepancies between the different copies of the sensors is of a mere 0.4 ± 0.3 µg.m$^{-3}$ [16]. Then, results from different Pollutrack sensors can be compared all together to study local variabilities.

An inter-comparison was previously conducted during several months in 2018 with Pollutrack sensors in parallel with 3 reference stations in the Paris region, France, from the air quality network Airparif [16]. The measurements were taken during different weather conditions and covered different levels of PM pollution (close to a motorway, in the centre of Paris, and in a small city). The cross-comparison analysis has shown an average discrepancy of 3 µg.m$^{-3}$ for daily measurements for all conditions, assuming no error in the reference measurements (it should be noted that the microbalance instruments usually used as reference could also have uncertainties of the same order [23]). Additionally, the mean uncertainty decreases when measurements are integrated over a period of several weeks or more, down to 0.1 µg.m$^{-3}$ [16] for both fixed and mobile stations.

Figure 3 presents the UBA reference measurements, the average measurements from the 4 fixed Pollutrack stations, and the modelling results. The Pollutrack sensors have measured the same time evolution of the PM2.5 as the UBA reference station for the 2021–2023 period, with a mean difference between the two datasets of 0.7 µg.m$^{-3}$ and a standard deviation of 2.3 µg.m$^{-3}$. The mean wind directions are mainly between the northwest and southwest; thus, the Pollutrack fixed stations and the UBA reference station are in the same air masses globally. Between the two measuring points are some commercial areas with industrial and logistics companies and highways, but their contribution is probably diluted during the air transport. The PM2.5 time evolution of the different sensors and the modelling calculations are also in excellent agreement for the year 2022, confirming the self-consistency of all the results when considering the measurement uncertainties. Figure 4 presents the histograms of the differences between the various sets of data, and Table 2 presents the main statistics. The histograms have a Gaussian shape with a centre relatively close to zero. The dispersion between Pollutrack and the reference data is low. The dispersion is higher but acceptable when considering the modelling data, confirming that the model is less accurate than real measurements.

**Table 2.** Mean values of the difference, standard deviations, and full width of half maximum values (FWHM) for the differences between the various datasets.

| Source of Values | Mean (µg.m$^{-3}$) | Standard Deviation (µg.m$^{-3}$) | FWHM (µg.m$^{-3}$) |
|---|---|---|---|
| Pollutrack—UBA reference | 0.7 | 2.3 | 2.2 |
| Modelling—UBA reference | 1.7 | 3.7 | 6.5 |
| Pollutrack—Modelling | −0.7 | 3.6 | 4.5 |

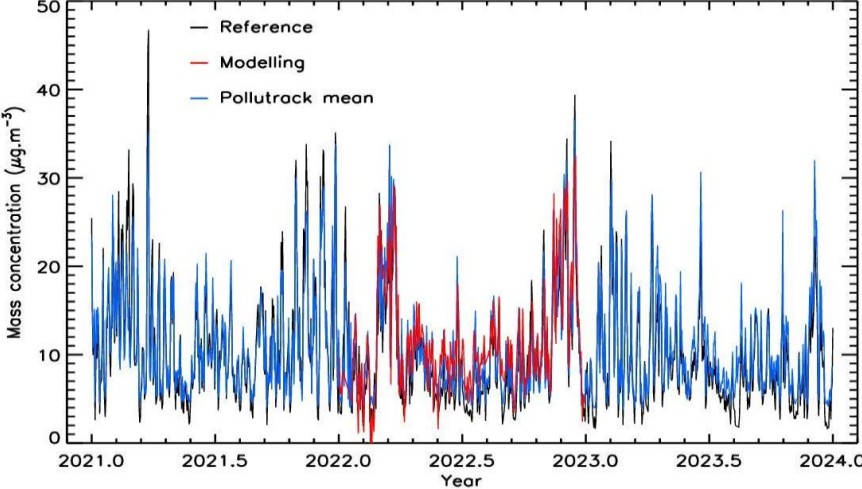

**Figure 3.** The time evolution of the PM2.5 concentrations for the UBA reference station, the Pollutrack fixed stations, and the modelling calculation.

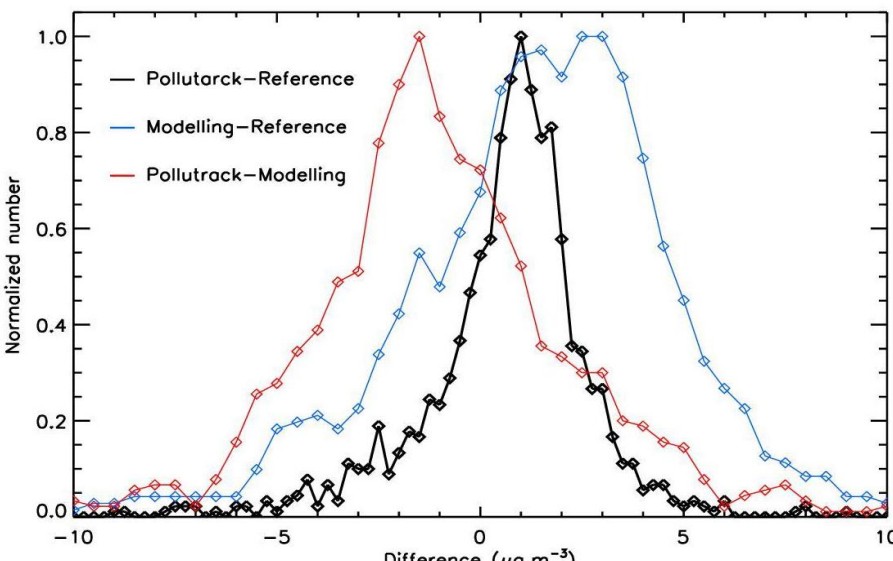

**Figure 4.** Histograms of the difference for the cross-comparison sessions of measurements (a sliding smoothing over 3 consecutive points is applied for the differences involving the modelling data due to a lower number of datapoints).

These results confirm the ability of Pollutrack mobile sensors to map the PM2.5 of Teltow district. The mean value of PM2.5 is 10.9 $\mu$g.m$^{-3}$ for Pollutrack sensors and 10.2 $\mu$g.m$^{-3}$ for the reference station at Blankenfelde-Mahlow station for the 2021–2023 period, with only a few pollution peaks above 30 $\mu$g.m$^{-3}$. These values are relatively low, indicating that the mean air quality was quite acceptable in this district during that period.

## 3. Results

The Pollutrack mobile sensors were mounted on two different types of vehicles, the roofs of cars that performed sparse measurements while travelling on different roads around Teltow and up to Potsdam at the west of Teltow and the roofs of buses that travelled on regular routes. The data, when available, are integrated on a square of 100 m side length. Figure 5 presents the locations of the measurements; the black dots represent the sparse measurements from cars and the red ones represent the regular measurements from buses. The concept of random measurements to produce daily or yearly maps of PM2.5 pollution works well in the case of a large number of vehicles with sensors, typically one hundred or more depending on the town size, as performed in other European cities. On the other hand, when the number of cars is too low, as for Teltow, although their travel covers a large part of the district, only some sporadic events could be accidentally detected. The number of measurements in a given location appears too low to apply statistics for producing the spatial PM2.5 maps. Thus, using the roof of buses is more appropriate for studies in midsize cities. Also, trucks for garbage collection could be considered in the future for that purpose, as currently experienced in Lille, northern France. Thus, we will consider the regular measurements only.

Figure 6 presents PM2.5 pollution maps for 2023 when averaging all values from regular measurements when available in the 100 m squares. At least 15,000 measurements per square are used. The mean lowest value is slightly below 11.5 $\mu$g.m$^{-3}$, above the recommended WHO annual mean (5 $\mu$g.m$^{-3}$) and above the future law-regulated threshold of the European Commission for 2030 (10 $\mu$g.m$^{-3}$). The mean value for the studied roads is 12.6 $\mu$g.m$^{-3}$, and the higher mean values are slightly above 14 $\mu$g.m$^{-3}$. A significant variability of about 25% is detected in the Teltow district, indicating some spatial heterogeneity for the local sources.

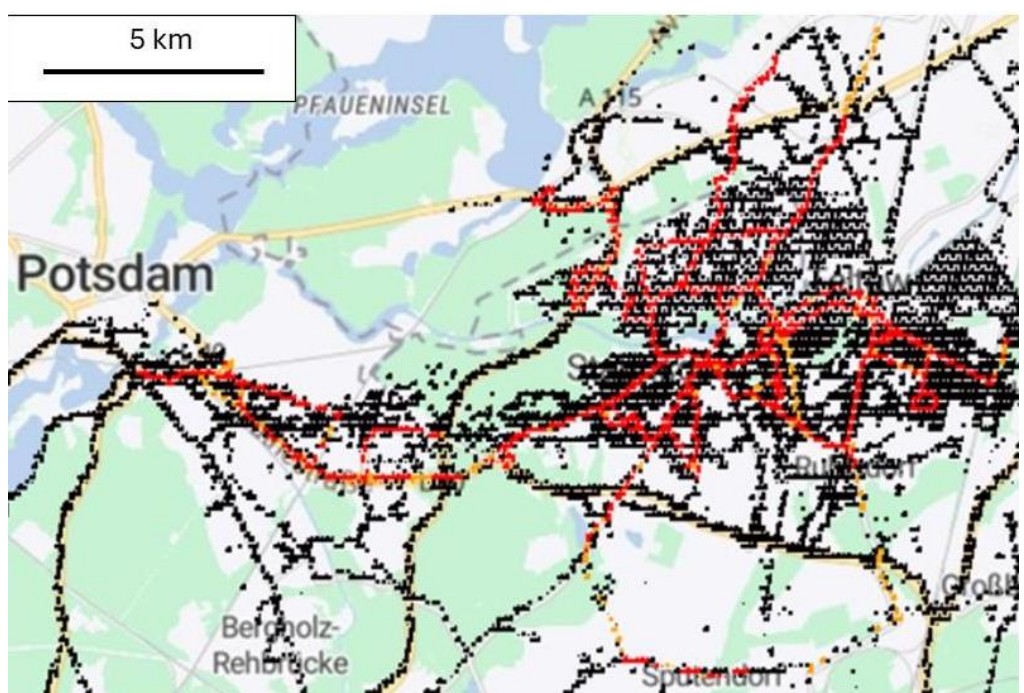

**Figure 5.** Locations of the measurements; black dots: sparse measurements; red dots: regular measurements.

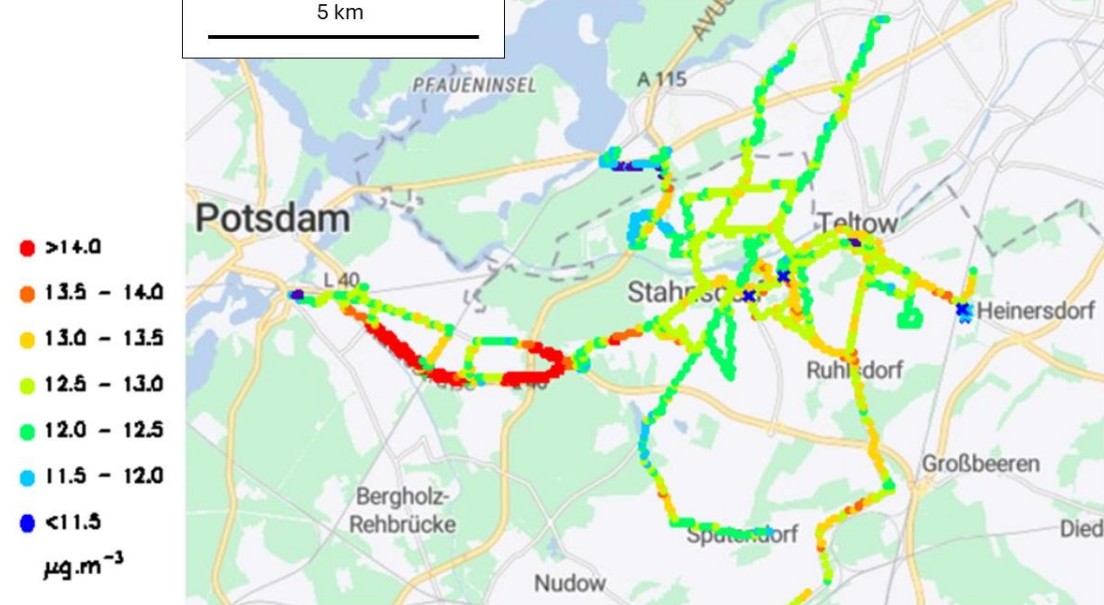

**Figure 6.** Mean PM2.5 mass concentrations from mobile sensors (thick square) and fixed sensors (thick crosses) superimposed on main roads in pale orange and secondary roads in pale grey.

The higher values are detected for the main roads and motorways, while the lower values are inside the centre of Teltow and in residential areas where the traffic is lower. Also, some very localised hot spots are present all over the city, perhaps due to red lights at the intersections.

The mean value from four Pollutrack fixed stations close to the Heinersdorf city is 10.3 $\mu$g.m$^{-3}$ for 2023. This value is almost 30% lower than the values on the highway, which are usually around 13.0 $\mu$g.m$^{-3}$. The latter is thus well above the average values of the UBA reference station, at 8.7 $\mu$g.m$^{-3}$ for 2023.

Two additional fixed stations have also provided continuous measurements since the beginning of 2023 in zones covered by mobile sensors. These stations are between the Teltow and Stahnsdorf cities, in moderately polluted zones between 10.4 and 12.0 $\mu$g.m$^{-3}$. The fixed stations are at least 2 m above ground and on the facade of a house or poles on the sidewalks (Figure 7). As for the four other fixed stations, the mean values are lower than those of the closest mobile measurements. When considering the six sets of measurements, a mean difference of $-1.9 \pm 0.7$ $\mu$g.m$^{-3}$ is found between fixed and mobile stations. This difference, above the measurement uncertainties, means that the values inside the traffic are significantly higher than those on the sidewalk of the streets. This could be due to the proximity of the mobile measurements to the emission sources (engines, brakes, tyres, and resuspended dust). The transport of particles is governed by the balance between gravity and air movements at the local scale. The PM concentration could be already diluted when reaching sidewalks. Thus, the air inhaled by car drivers and bicycle drivers on the streets could be more polluted, by about 14% for PM2.5 mass concentrations, than the air inhaled by pedestrians. Of course, these results must be confirmed and refined by new sessions of measurements involving more fixed stations.

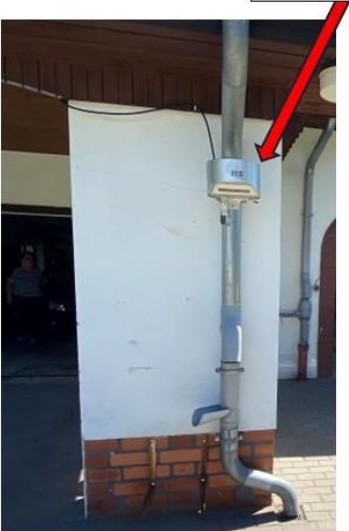 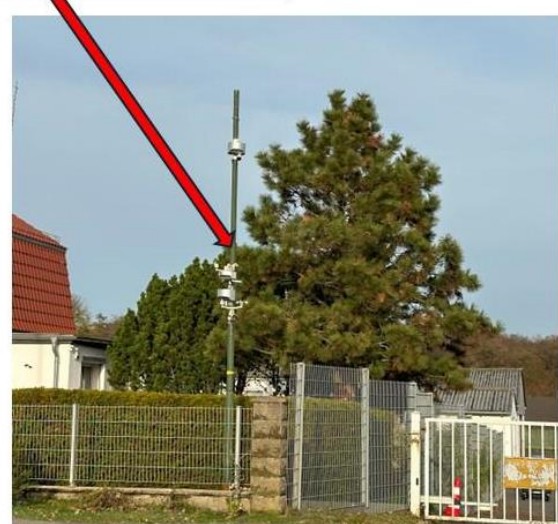

**Figure 7.** The installation of the fixed sensors.

The second parameter to evaluate the sanitary risk of PM2.5 exposure is the number of days per year above the 15 $\mu$g.m$^{-3}$ daily ceiling that the WHO recommends not to exceed more than four days per year. Figure 8 shows that the highest number of days occurs where the mean mass concentrations are higher, on the main roads and motorways, and for the hot spots inside the city. For the mean of the fixed Pollutrack stations and the UBA reference station, the number of days per year is 75 and 65, respectively. As with mass concentration data, these values are below the values from mobile sensors, where 90 days is the lowest value detected in the city. The highest number of days, above 130, is recorded on the southwest motorway with levels up to 40% higher than in the centre of Teltow city. Il should be noted that there is no direct residential development near this highway.

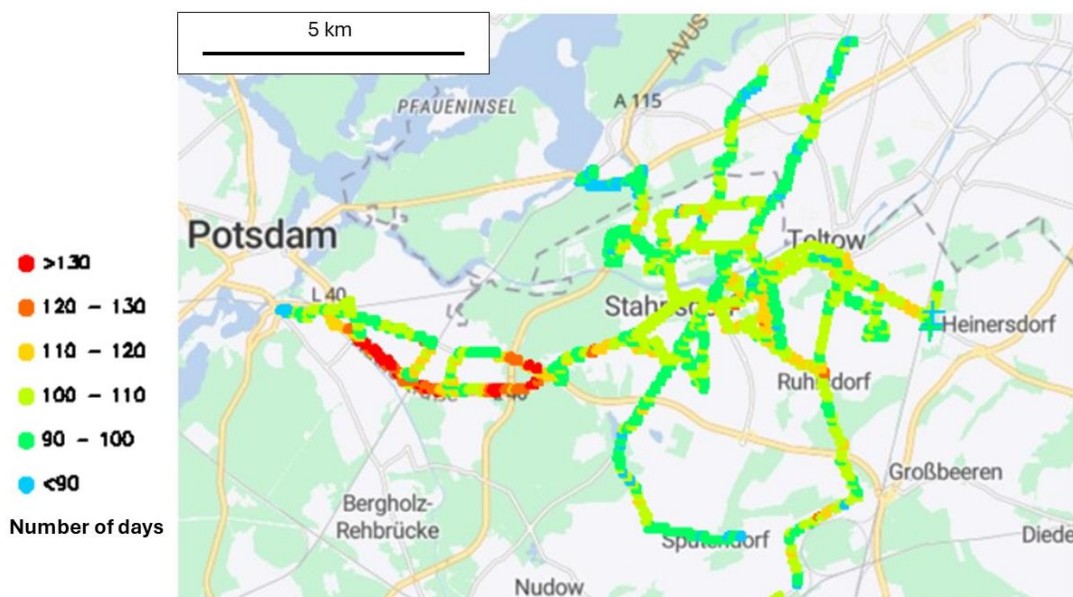

**Figure 8.** The mean number of days with PM2.5 mass concentrations above 15 μg.m$^{-3}$ from the mobile sensors (thick square) superimposed on main roads in pale orange and secondary roads in pale grey.

## 4. Discussion

The sources of pollution can be multiple for primary and secondary PM2.5 particles. The mean mass concentration values result from mixtures of motor exhausts, mainly on motorways and highways [23], of car brakes and tyre abrasion on all roads, resuspended dust, industrial and agricultural activities, domestic combustion, and transported particles over long distances [24–27]. It is reasonable to assume that the higher values are dominated by the local traffic and sometimes, depending on the wind direction and strength, by local industrial activities. Better identifying the nature, origin, and transport at the local scale of PM will be of high importance, as studies suggest a strong link between human health and carbonaceous PM [28], while the effect of metal ones must also be considered [29,30].

Paris, France, is the only city where similar studies were published, using the Pollutrack sensors to provide maps with a resolution of 1 km over 5 years [16]. Because Paris is one of the most polluted cities in Western Europe due to the traffic and the local topography with hills and canyon streets, the mean mass concentration values are higher than in Teltow by up to 40% (14 vs. 19 μg.m$^{-3}$ for the most polluted parts of the cities). The highest values are recorded on the motorway that encircles Paris, with values 60% higher than those of the less polluted parts of the city. The analogy indicates that motorways and highways correlate more clearly with traffic as a source of PM than urban centres. This is due to emissions and continuous turbulences that produce resuspension of the PM. It is also possible to assume cumulative measurements, i.e., the same particles are measured repeatedly, as they cannot be deposited on the road surface but are whirled up.

Daily reference values provided by air quality networks can sometimes be underestimated if the hot spots and the most popular streets are not accurately documented. The results show that a midsize city like Teltow can have the same situation as larger cities when highways and urban motorways are present. All this argues in favour of establishing PM2.5 pollution maps with a high spatial resolution to accurately determine the real hyper-local pollution levels.

Mobile sensors on buses or trucks for garbage collection, coupled with fixed stations, can be a good solution to establish accurate emissions registers. Identifying hot spots is of strong interest to tentatively reduce PM emissions and to better consider the utmost critical street ventilation for future urban plans. Also, citizens should use real-time pollution maps to avoid intense sport activities in the most polluted areas of the cities and in urban

peripheral areas. Since the local recreation areas and areas for sporting activities are often located in the urban fringe areas, expressways and highways in these urban fringe areas are particularly important places to monitor in the future. Timely information about air quality for citizens can significantly improve health with simple instructions, such as "Recommendation of limited activities in highly polluted areas today". Finally, such permanent measurements with high spatial resolution should be used in the future by real-time modelling calculations and assimilation processes to readjust the modelling outputs and produce global realistic maps, even for locations without measurements.

### 5. Conclusions

Fixed and mobile Pollutrack sensors are installed in over 30 large European cities (www.pollutrack.net, accessed on 29 October 2024). The first steps of measurement analysis presented here for Teltow district and previously for Paris should be applied to other cities with specificities. Some are relatively well ventilated (as in Teltow, in a relatively flat landscape), while others are not (as in Paris, surrounded by hills). Some cities are close to the sea and can encounter episodes of marine pollution depending on the wind direction (like in Dublin, Ireland, or Marseille, France), and some are in industrial regions (such as in Warsaw, Poland). Close collaboration with air quality research teams and agencies that develop and operate reference stations and modelling works on PM pollution must be conducted in the future to incorporate these high-spatial resolution measurements with their real-time analysis capacity.

Although PM2.5 mass concentration measurements are a useful parameter to evaluate pollution, information on the size distribution and the nature of particles is missing. Such parameters are necessary to better evaluate the complex origins and the transport of particles. For size distributions, the Pollutrack sensors that also provide number concentrations for particles greater than 0.5 μm could be used [16]. For more accurate size distribution measurements from 0.2 μm to 100 μm and for an estimation of particle typology (liquid, black carbon, organic carbons, mineral salt, metals), more expensive aerosol counters are required [31]. Also, such measurements could be combined with instrumentations that provide in real time the chemical composition of the particles, such as an aerosol chemical speciation monitor and aethalometer [32].

This multimodal approach, combining fixed reference stations, accurate mobile sensors, and data assimilation models, offers the most effective solution for precisely assessing citizens' real-time exposure to PM2.5 throughout the day, as well as understanding the nature of the particles they inhale. By providing real-time air quality data and public education about pollution levels, citizens, especially pedestrians and cyclists, can make informed choices about routes with lower pollution levels, thereby minimizing health risks. In highly polluted areas, wearing an FFP2 mask and limiting physical activity, particularly for individuals with pre-existing health conditions, can further reduce exposure. To achieve this, daily street-level PM2.5 pollution maps, based on real measurements rather than solely on modelling calculations, should be made readily accessible to all citizens.

The Pollutrack sensors continue to operate in Teltow, enabling analysis to extend through 2025 to confirm the persistence of the identified hot spots and examine their seasonal variability. Also, new Pollutrack fixed stations operational since the end of 2023 are being used to better evaluate the difference between PM values inside the traffic and on the sidewalks.

In our ongoing research, we are utilizing the DEUS modular system, which features an AI-powered camera (DEUS Ai-Vision) for intelligent traffic detection, providing real-time, reliable data on the number and categories of road users. Complementing this, the DEUS multi-gas module monitors the concentrations of $NO_2$, $CO_2$, and VOCs in targeted areas. In an upcoming paper, we will present an in-depth analysis of the temporal correlation between traffic patterns, $NO_2$ levels, and PM2.5 concentrations on a local scale, offering valuable insights into the contribution of local traffic to overall air pollution.

**Author Contributions:** Conceptualization, J.-B.R., G.B., M.N., E.T., L.T., E.P., and M.S.; methodology, J.-B.R.; software, J.-B.R.; validation, J.-B.R.; formal analysis, J.-B.R.; investigation, J.-B.R., G.B., M.N., E.T., L.T., E.P, M.S., and J.S.; resources, M.N. and E.P.; data curation, M.S., M.N., and E.P.; writing—original draft preparation, J.-B.R.; writing—review and editing, G.B., M.N., E.T., L.T., E.P., M.S., and J.S.; visualization, J.-B.R.; supervision, M.N. and E.P.; project administration, M.N. and E.P.; funding acquisition, M.N. and E.P. All authors have read and agreed to the published version of the manuscript.

**Funding:** This work is partially funded through the mFUND projects "DEUS-SmartAir" and "KoFeMo" by the German Federal Ministry for Digital and Transport Affairs (BMDV).

**Data Availability Statement:** The data presented in this study are available on request from the Pollutrack and DEUS Company who own them.

**Acknowledgments:** We extend our sincere gratitude to all the employees of the city of Teltow, with special thanks to Thomas Schmidt for their invaluable collaboration and support throughout this study. We also wish to acknowledge Regiobus Potsdam Mittelmark GmbH, McDonald's Teltow, and Zweckverband Bauhof TKS for generously providing their fleets to carry our mobile sensors, as well as Scott Stonham for his most efficient proofreading. Furthermore, we are grateful for the support and resources from the German Federal Ministry for Digital Affairs and Transport, made available through the mFUND innovation initiative, as part of the KoFeMo and DEUS Smart Air projects, which made this research possible.

**Conflicts of Interest:** The authors declare no conflicts of interest.

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
