# Peer review of "High-Spatial Resolution Maps of PM2.5 Using Mobile Sensors on Buses: A Case Study of Teltow City, Germany, in the Suburb of Berlin, 2023"

_atmosphere, doi:10.3390/atmos15121494_

Round 1

Reviewer 1 Report

Comments and Suggestions for Authors

Dear Authors,

Thank you for the opportunity to review your manuscript titled "High spatial resolution maps of PM2.5 using mobile sensors on buses: the case study of Teltow city, Germany, in the suburb of Berlin, for 2023." I appreciate the innovative approach taken in monitoring air quality using mobile sensors and the integration of high spatial resolution mapping for PM2.5 pollution exposure analysis. Below are my suggestions to enhance the clarity and impact of the paper:

  1. English Language and Grammar:
    • The manuscript would benefit from a thorough review of English grammar and sentence structure to improve readability. Some sentences are complex or unclear, and refining the language will help in better conveying the scientific findings to the audience.
  2. Conciseness:
    • Please aim to make the abstract and certain sections of the paper more concise. Reducing redundant phrases and maintaining a focused discussion will improve the flow and accessibility of your study’s key points.
  3. Details on Equipment and Sensors:
    • It would be valuable to provide more information regarding the specific equipment and sensors installed on the buses. Please include images or figures that depict the system setup on the buses, as well as a flow diagram of how data is collected, transferred, and processed. This will offer readers a clearer understanding of the technical setup and data acquisition process.
  4. References:
    • Additional references to support your methodology and findings would strengthen the paper. It would be helpful to include prior studies that have used similar mobile sensing techniques or high-resolution air quality mapping to add context to your work.
  5. Technical Details of Pollutrack Sensors:
    • Further details on the Pollutrack SAS particulate matter sensors are recommended. This could include specifics on their sensitivity, accuracy, and any calibration methods used to ensure data reliability. If applicable, a comparison with other sensor types may be beneficial for readers unfamiliar with Pollutrack technology.
  6. Comparison with Other Locations:
    • While you mention an analogy with results from Paris, France, providing additional quantitative comparisons (e.g., PM2.5 levels, traffic density) between Teltow and Paris would add depth to your discussion and emphasize the broader applicability of your methodology.
  7. Data Consistency and Validation:
    • Please elaborate on how data consistency and validation were achieved. Highlight any cross-referencing methods used with German Environment Agency (UBA) measurements or model calculations, to ensure readers are confident in the robustness of the data.
  8. Environmental Impact Discussion:
    • Consider expanding the discussion on the implications of PM2.5 hotspots along high-traffic roads and motorways. Exploring the potential health impacts on pedestrians and commuters in these areas, or proposing mitigation strategies, could add value to your conclusions.
  9. Future Applications and Improvements:
    • A brief section on possible improvements to the system or its application in other urban environments would enhance the paper's contribution to the field. Discussing future work could also provide guidance for researchers interested in replicating or building upon your study.

Best regards

Comments on the Quality of English Language

Need supports

Author Response

Review Report 1

Thank you for the opportunity to review your manuscript titled "High spatial resolution maps of PM2.5 using mobile sensors on buses: the case study of Teltow city, Germany, in the suburb of Berlin, for 2023." I appreciate the innovative approach taken in monitoring air quality using mobile sensors and the integration of high spatial resolution mapping for PM2.5 pollution exposure analysis. Below are my suggestions to enhance the clarity and impact of the paper:

Authors’ answer: We thank the reviewer for their comments that have help us to improve the paper.

  1. English Language and Grammar:
    • The manuscript would benefit from a thorough review of English grammar and sentence structure to improve readability. Some sentences are complex or unclear, and refining the language will help in better conveying the scientific findings to the audience.

Authors’ answer: We agree that some sentences were too complex. We have tried to improve the language.

  1. Conciseness:
    • Please aim to make the abstract and certain sections of the paper more concise. Reducing redundant phrases and maintaining a focused discussion will improve the flow and accessibility of your study’s key points.

Authors’ answer: The abstract is shortened. We have deleted several redundant sentences. We have tried to improve the flow of the discussion

  1. Details on Equipment and Sensors:
    • It would be valuable to provide more information regarding the specific equipment and sensors installed on the buses. Please include images or figures that depict the system setup on the buses, as well as a flow diagram of how data is collected, transferred, and processed. This will offer readers a clearer understanding of the technical setup and data acquisition process.

Authors’ answer:

We have added in the text:

“The shape of the instrument was especially designed, with the inlet at the opposite of the vehicle motion (Figure 2).”

“Figure 2: Pollutrack sensors (inside the red circle) on the roof of a bus (left) and of a car (right).

 “The data are processed onboard the sensors to provide number and mass-concentrations. They are sent in real time through a mobile phone system to the Pollutrack data center where they are stored. The data are then reprocessed, taking into account the humidity values obtained from the nearest weather station. Finally, the data are averaged over 30 minutes.“

“For such application, time, GPS position and Pollutrack measurements are available every 10 seconds. As for fixed stations, the data are sent through mobile phone system to the Pollutrack data center and are reprocessed using humidity data from the closest weather station. Finally, the data are hourly averaged for a square of 100 m in edge length.”

  1. References:
    • Additional references to support your methodology and findings would strengthen the paper. It would be helpful to include prior studies that have used similar mobile sensing techniques or high-resolution air quality mapping to add context to your work.

Authors’ answer: References to methodology and results were already given (references 15 and 16). To our knowledge, no other similar works have been published. But we understand the reviewer concern. Thus, we have added in the introduction:

“at a scale of a kilometer can occur [15,16].”

“While the maps of PM2.5 for Paris has a spatial resolution of 1 km, we propose here to study the pollution at the street scale.”

  1. Technical Details of Pollutrack Sensors:
    • Further details on the Pollutrack SAS particulate matter sensors are recommended. This could include specifics on their sensitivity, accuracy, and any calibration methods used to ensure data reliability. If applicable, a comparison with other sensor types may be beneficial for readers unfamiliar with Pollutrack technology.

Authors’ answer: Such work was already published, and the results were given at the beginning of part 2.3. We agree that a longer summary of these previous work should be provided. We have added in the text:

“An inter-comparison was previously conducted during several months in 2018 with Pollutrack sensors in parallel with 3 reference station in the Paris region, France, from the air quality network Airparif. The measurements were taken during different weather conditions and covered different levels of PM pollution (close to a motorway, in the center of Paris and in a small city). The cross-comparison analysis has shown an average discrepancy of 3 µg.m-3 for daily measurements for all conditions”

  1. Comparison with Other Locations:
    • While you mention an analogy with results from Paris, France, providing additional quantitative comparisons (e.g., PM2.5 levels, traffic density) between Teltow and Paris would add depth to your discussion and emphasize the broader applicability of your methodology.

Authors’ answer: We have already given the difference between Paris and Teltow in %. We have now added the values “(14 vs. 19 µg.m-3 for the most polluted parts of the cities)”. No traffic data were available in Teltow at the time of this study. Traffic counting system are now installed and will be used in a future paper on the Teltow pollution.

  1. Data Consistency and Validation:
    • Please elaborate on how data consistency and validation were achieved. Highlight any cross-referencing methods used with German Environment Agency (UBA) measurements or model calculations, to ensure readers are confident in the robustness of the data.

Authors’ answer: How the Pollutrack sensors has been validated is given at the beginning of part 2.3 (see comment #5). The calculation of the mean difference and standard deviation between two data sets is the best method to evaluate the self-consistency of the measurements, as already explained at the end of section 2.3 and presented in Table 2. We have added in the text:

“, with a mean difference between the two data sets of 0.7 µg.m-3 and a standard deviation of 2.3 µg.m-3.”

  1. Environmental Impact Discussion:
    • Consider expanding the discussion on the implications of PM2.5 hotspots along high-traffic roads and motorways. Exploring the potential health impacts on pedestrians and commuters in these areas, or proposing mitigation strategies, could add value to your conclusions.

Authors’ answer: Fortunately, there is no pedestrian on the higher pollution zone that are located on the high-roads. We have already discussed this impact in the paper, but we have re-organize some parts of the discussion.

We have added in the text: “For that purpose, daily maps of PM2.5 pollution map at the street scale constrained by real measurements and not only for modelling calculations. should be available to all citizens.”

“It should be noted that there is no direct residential development near this highway.”

“Better identifying the nature, the origin, and the transport at the local scale of PM will be of high importance as studies suggest a strong link between human health and carbonaceous PM”

We have rewritten the third paragraph of the conclusion:” This multimodal approach, combining fixed reference stations, accurate mobile sensors, and data assimilation models, offers the most effective solution for precisely assessing citizens' real-time exposure to PM2.5 throughout the day, as well as understanding the nature of the particles they inhale. By providing real-time air quality data and public education about pollution levels, citizens, especially pedestrians and cyclists, can make informed choices about routes with lower pollution levels, thereby minimizing health risks. In highly polluted areas, wearing an FFP2 mask and limiting physical activity, particularly for individuals with pre-existing health conditions, can further reduce exposure. To achieve this, daily street-level PM2.5 pollution maps, based on real measurements rather than solely on modeling calculations, should be made readily accessible to all citizens.”

  1. Future Applications and Improvements:
    • A brief section on possible improvements to the system or its application in other urban environments would enhance the paper's contribution to the field. Discussing future work could also provide guidance for researchers interested in replicating or building upon your study.

Authors’ answer: We have already answered to these comments in the paper: “Although PM2.5 mass-concentration measurements are a useful parameter to evaluate pollution, information on the size distribution and the nature of particles are missing. Such parameters are necessary to better evaluate the complex origins and the transport of particles. For size distributions, the Pollutrack sensors that also provide number concentrations for particles greater than 0.5µm could be used [16]. For more accurate size distribution measurements from 0.2 µm to 50 µm and for estimation of particle typology (liquid, black carbon, organic carbons, mineral salt, metals), more expensive aerosol counters should be used [31]. Also, such measurements could be combined with instrumentations that provide in real time the chemical composition of the particles, as Aerosol Chemical Speciation Monitor and Aethalometer [32].”

We rewritten the end of the conclusion:” through 2025 to confirm the persistence of the identified hot spots and examine their seasonal variability. Also, new Pollutrack fixed stations operational since the end of 2023 are being used to better evaluate the difference between PM values inside the traffic and on the sidewalks.

In our ongoing research, we are utilizing the DEUS modular system, which features an AI-powered camera (DEUS Ai-Vision) for intelligent traffic detection, providing re-al-time, reliable data on the number and categories of road users. Complementing this, the DEUS multi-gas module monitors concentrations of NOâ‚‚, COâ‚‚, and VOCs in targeted are-as. In an upcoming paper, we will present an in-depth analysis of the temporal correlation between traffic patterns, NOâ‚‚ levels, and PM2.5 concentrations on a local scale, offering valuable insights into the contribution of local traffic to overall air pollution.”.

Reviewer 2 Report

Comments and Suggestions for Authors

The paper presents initial results of an interesting mobile monitoring study of PM, performed in the suburb of Berlin. It could be interesting for the readers of Atmosphere, but only after following changes are made.

Major:

In Section 2.1 give the details of the used sensors, and air sampling details (what kind of inlet was used in fixed vs mobile setting, what was the air volumetric flow rate, and how was the correction done for the speed of the vehicle in mobile measurements).

Improve the quality of Figure 2, since the differences between different signals are not cleary visible

In Table 2, please add more descriptive statistics about Pollutrack, Modelling and UBA references (such as percentiles). Also add a plot of distribution of these 3 variables. Comment on the newly added results.  

Figure 4, improve resolution of the figure. Also be more precise about the way in which the mean was calculated. Also calculate the median, and describe if there was a presence of outliers, and how was this handeled. 

Add a plot with similar color scheme with data coverage (instead of concentration, ploted value is how much data points are gathered per road segment (dot in the Figure)). Similar improvements should be made for Figure 6.

Much more details should be added about Pollutrack sensors used (sensor suite/type of sensors present in nodes, air sampling for fixed and mobile monitoring). Furthermore, uncertainty budget should be estimated for the two methodologies. 

Minor: 

Line 18: "exposure to pollution", reconsider replacing with "exposure to outdoor air pollution"

Line 18-20: This is not a suitable place to state funding sources, please move to the dedicated section at the end of the manuscript.

Line 21-23: It is not entirely clear what was installed and where, please rephrase to make it more clear.

Line 29: Add the year for the new WHO limits.

Line 30: Also give here the EU limits, since these are different than WHO limits.

Line 55-56: Consider being more precise when talking about outdoor and indoor air pollution (which would be present in "living places").

Line 91-92: Rephrase "inside and outside the traffic", since it is not clear, weather the authors are reffering to rush hours, or streets vs sidewalk, or strets vs green areas.

Line 194: Why use "square of 100m side length" when a road section of certain lenght is shown later in the paper? 

Line 273: "The only city where similar studies were conducted is Paris, France, with a spatial resolution of 1 km"

Mobile monitoring was performed in a number of different cities, using both lab grade and LCS instruments, please rephrase, and add additional references. 

Comments on the Quality of English Language

English language:

Line 44: "they have" it should be "it has" (particulate matter), subject-verb agreement: "PM2.5 has" instead of "they have"

Line 66: Use more widely used term instead of "one-to-one regulation".

Line 77: "On the other range" please rephrase

Line 91-92: "within situ", is it "with in-situ"? 

Line 109: "such a campaign should show an interest", please rephrase

Line 191: "non-reproductive" please use a better phrase

Line 201: "could be fortunately detected" better to use different phrasing e.g. "serendipitously"

Please consult native speaker to improve the overall quality of language, since the above list of omissions is not exhaustive.

Author Response

Review Report 2

Comments and Suggestions for Authors

- The paper presents initial results of an interesting mobile monitoring study of PM, performed in the suburb of Berlin. It could be interesting for the readers of Atmosphere, but only after following changes are made.

Authors’ answer: We thank the reviewer for their comments that have help us to improve the paper.

- In Section 2.1 give the details of the used sensors, and air sampling details (what kind of inlet was used in fixed vs mobile setting, what was the air volumetric flow rate, and how was the correction done for the speed of the vehicle in mobile measurements).

Authors’ answer: Some information are on the property of the Pollutrack society and cannot be disclose. The inlet is the same for both instruments, but the package is different.

We have added in the text for the mobile instrument: “The flow is 0.2 litter per minute “. “The shape of the instrument was then specially designed, with the inlet at the opposite of the vehicle motion (Figure 2).”

- Improve the quality of Figure 2, since the differences between different signals are not cleary visible

Authors’ answer: The difference are difficult to see because the curves are very similar. We have tried to improve the figure (now figure 3) by reducing the y-axis range and by removing the dots.

- In Table 2, please add more descriptive statistics about Pollutrack, Modelling and UBA references (such as percentiles). Also add a plot of distribution of these 3 variables. Comment on the newly added results.  

Authors’ answer: We have added a new figure (Figure4):

“Figure 4. Histograms of the difference for the cross-comparison sessions of measurements (a sliding smoothing over 3 consecutive points is applied for the differences involving the modelling data due to a lower number of data).”

We have added in the text: “Figure 4 presents the histograms of the differences between the various sets of data, and Table 2 presents the main statistics. The histograms have a Gaussian shape with a center relatively close to zero. The dispersion between the Pollutrack and the reference data is low. The dispersion is higher but acceptable when considering the modelling data, confirming that the model is less accurate than real measurements..”

Finally, we have added in the table 2 the full width of half maximum values (FWHM) for the histograms. We have changed the legend to: ‘Table 2. Mean values or the difference, standard deviations and full width of half maximum values (FWHM) for the differences between the various data sets.”

- Figure 4, improve resolution of the figure. Also be more precise about the way in which the mean was calculated. Also calculate the median, and describe if there was a presence of outliers, and how was this handeled. 

Authors’ answer: In our version, the resolution seems good. Nevertheless, we have enlarge the figure (now figure 6).

The mean is calculated by adding the measurements at a given location and dividing by the number of measurements. No outliers are present in the data (in fact, if outliers exist, they are automatically remove by the Pollutrack software).

We have added in the text:” Figure 6 presents PM2.5 pollution maps for 2023 when averaging all values from regular measurements when available in the 100 m squares.”

We have considered mean values instead of median values, because the WHO recommendations, as well as the regulation monitoring values, consider mean values. If we want to compare our results to such references, we must calculate the means.

- Add a plot with similar color scheme with data coverage (instead of concentration, ploted value is how much data points are gathered per road segment (dot in the Figure)). Similar improvements should be made for Figure 6.

Authors’ answer: We are not sure that such figures are necessary.

We have added in the text:” At least 15 000 measurements per square are used.”

- Much more details should be added about Pollutrack sensors used (sensor suite/type of sensors present in nodes, air sampling for fixed and mobile monitoring). Furthermore, uncertainty budget should be estimated for the two methodologies. 

Authors’ answer:  The sensors are provided by the Pollutrack company. Previous studies have been conducted concerning the measurements errors.

We have added: “The Pollutrack sensors, developed and provided by the Pollutrack company (France),“

”The flow is 0.2 litter per minute.”

 “An inter-comparison was previously conducted during several months in 2018 with Pollutrack sensors in parallel with 3 reference station in the Paris region, France, from the air quality network Airparif. The measurements were conducted during different weather conditions and to cover different situations of PM pollution (close to a motorway, in the center of Paris and in a small city. The humidity conditions retrieved from the nearest weather station are including in in the data processing software. The ross-comparison analysis has shown an average discrepancy of 3 µg.m-3 for daily measurements for all conditions,”

“the mean uncertainties decrease when measurements are integrated over a period of several weeks or more, down to 0.1 µg.m-3 [16] for both fixed and mobile stations.”

Minor: 

- Line 18: "exposure to pollution", reconsider replacing with "exposure to outdoor air pollution"

Authors’ answer:  Done.

- Line 18-20: This is not a suitable place to state funding sources, please move to the dedicated section at the end of the manuscript.

Authors’ answer:  The sentence is removed.

- Line 21-23: It is not entirely clear what was installed and where, please rephrase to make it more clear.

Authors’ answer:  Indeed, this sentence was unclear. We have changed to: “A network of optical sensors from the Pollutrack Company was established with fixed stations and sensors on the roofs of buses and cars,”

- Line 29: Add the year for the new WHO limits.

Authors’ answer:  We have added:” in 2021”

- Line 30: Also give here the EU limits, since these are different than WHO limits.

Authors’ answer: We have removed the “EU limits” since we do not speak of them in the text.

- Line 55-56: Consider being more precise when talking about outdoor and indoor air pollution (which would be present in "living places").

Authors’ answer: Our aim was not to speak about indoor air pollution. We have changed the text to “outdoor activities”.

- Line 91-92: Rephrase "inside and outside the traffic", since it is not clear, weather the authors are reffering to rush hours, or streets vs sidewalk, or strets vs green areas.

Authors’ answer: We have changed the text to “ inside the traffic flow and on sidewalks”.

- Line 194: Why use "square of 100m side length" when a road section of certain lenght is shown later in the paper? 

Authors’ answer: The data from the mobile sensors are integrated and provided with the Polutrack/DEUS data system with this 100 m spatial resolution. For the fixed sensors, we consider the exact location. We can compared the two sets of data, even if the spatial resolution in not the same, considering that sources of the pollution detected by the mobile sensors are on the streets and not on the sidewalks.

- Line 273: "The only city where similar studies were conducted is Paris, France, with a spatial resolution of 1 km"

Mobile monitoring was performed in a number of different cities, using both lab grade and LCS instruments, please rephrase, and add additional references. 

Authors’ answer: We agree that some campaigns have been locally conducted with mobile sensors. Nevertheless,  PM2.5 monitoring with tens or hundreds of mobile sensors during several months or years are done only by the Pollutrack network.

We have changed the sentence to: “Paris, France, is the only city where similar studies were published using the Pollutrack sensors, to provide maps with a resolution of 1 km over 5 years”.

Comments on the Quality of English Language

English language:

-Line 44: "they have" it should be "it has" (particulate matter), subject-verb agreement: "PM2.5 has" instead of "they have"

Authors’ answer: Done.

- Line 66: Use more widely used term instead of "one-to-one regulation".

Authors’ answer: We have removed it.

- Line 77: "On the other range" please rephrase

Authors’ answer: We have removed it.

- Line 91-92: "within situ", is it "with in-situ"? 

Authors’ answer: Corrected.

- Line 109: "such a campaign should show an interest", please rephrase

Authors’ answer: We have changed the sentence to: ‘Compared to the measurements in the high-populated city of Paris, this new campaign should identify the presence of hot spots in a moderately populated city.’

- Line 191: "non-reproductive" please use a better phrase

Authors’ answer: We have removed this word.

- Line 201: "could be fortunately detected" better to use different phrasing e.g. "serendipitously"

Authors’ answer: Corrected.

- Please consult native speaker to improve the overall quality of language, since the above list of omissions is not exhaustive.

Authors’ answer: This is now done.

Reviewer 3 Report

Comments and Suggestions for Authors

The Article “High spatial resolution maps of PM2.5 using mobile sensors on buses: the case study of the Teltow city, Germany, in the suburb of Berlin, for 2023”, do describe an interesting methodology to monitor and extract air pollution data that can be mapped and thus further critically analysed for citizens, scientists and decision makers.

The text is comprehensive. My few comments are:

Some subscripts are not formatted as in NOx, etc.;

What about missing data due to “normal campaigns problems”?;

Table 2: “Pollutrack – Modelling” mean is negative. Explain better it;

Line 2011: “Figure 4 presents PM2.5 pollution maps for 2023”. This is the average of 365 days of 2023? And 24h/day of measurements? Or at night you had fewer measurements? Implications for computed averages in 100 mx100m?

Lines 278-283 do express important point but could be more clear on it. It is resuspension of particles (if dry weather) and bigger residence time of pollution species (mostly intense in days of low wind and/or due to nearby obstacles).

Line 292: “meshed” not the best word here;

Line 301: “No activities in area A or B today”. Could explain reason and thus be “No activities in area A or B today, due to bad air quality”.

Line 309: Teltow is well-ventilated compared to Paris means that Teltow has less buildings/obstacles or also that Teltow is windier (that region), yearly average?!;

With so much data, some more maps could be presented, also taking into account that the paper size is not high. Maybe some monthly average data (for 2 chosen months) could be interesting for readers.

Congratulations for all work but specially for the experimental part which is hard due to the articulation with all the volunteer institutions that cooperated with mobile sensors.

Author Response

Review Report 3

Comments and Suggestions for Authors

The Article “High spatial resolution maps of PM2.5 using mobile sensors on buses: the case study of the Teltow city, Germany, in the suburb of Berlin, for 2023”, do describe an interesting methodology to monitor and extract air pollution data that can be mapped and thus further critically analysed for citizens, scientists and decision makers.

Congratulations for all work but specially for the experimental part which is hard due to the articulation with all the volunteer institutions that cooperated with mobile sensors.

The text is comprehensive. My few comments are:

Authors’ answer: We thank the reviewer for their comments that have help us to improve the paper.

Some subscripts are not formatted as in NOx, etc.;

Authors’ answer: Corrected.

What about missing data due to “normal campaigns problems”?;

Authors’ answer: Of course, occasionally, some data can be erroneous, or lost during the transmission. Few erroneous data are removed by the Pollutrack network software. The sensors are very robust, as established during previous campaign since 2018 in other cities, and no mobile sensors were lost during the campaign.

Table 2: “Pollutrack – Modelling” mean is negative. Explain better it;

Authors’ answer: It is just because mean Pollutrack value is little smaller than the mean modelling value. We have added: “Mean values of the difference”.

Line 2011: “Figure 4 presents PM2.5 pollution maps for 2023”. This is the average of 365 days of 2023? And 24h/day of measurements? Or at night you had fewer measurements? Implications for computed averages in 100 mx100m?

Authors’ answer: The 2023 values are the mean of the daily values available in each square. At least 15000 of individual values are available in each square. Obviously, there as just few or no measurements during the night due to the bus operating hours during these hours. Since this situation occurs for all mobile sensors, we can compare the resulting means without bias.

Lines 278-283 do express important point but could be more clear on it. It is resuspension of particles (if dry weather) and bigger residence time of pollution species (mostly intense in days of low wind and/or due to nearby obstacles).

Authors’ answer: Our data do not allows to go further in detail on these processes. We have changed the sentence to: “This is due to emissions but also to continuous turbulences that produce resuspension of the PM.”

Line 292: “meshed” not the best word here;

Authors’ answer: We have changed to “coupled”.

Line 301: “No activities in area A or B today”. Could explain reason and thus be “No activities in area A or B today, due to bad air quality”.

Authors’ answer: We have changed the sentence to:” Recommendation of no activities in highly polluted areas today”.

Line 309: Teltow is well-ventilated compared to Paris means that Teltow has less buildings/obstacles or also that Teltow is windier (that region), yearly average?!;

Authors’ answer: Paris is surrounded by hills. Also, indeed, Teltow has less building and obstacles and is in a relatively flat landscape. We have changed the sentence to; “Some are relatively well-ventilated (as Teltow, in a relatively flat landscape), others are not (as Paris, surrounded by hills)”

With so much data, some more maps could be presented, also taking into account that the paper size is not high. Maybe some monthly average data (for 2 chosen months) could be interesting for readers.

Authors’ answer: We have also analyzed the data by months or during seasons, and nothing special appears for Teltow. Thus, it seems not necessary to add more figures. On a future work, as said in the conclusion, we will also consider the NO2 measurements and traffic counting.

Reviewer 4 Report

Comments and Suggestions for Authors

This paper yields an interesting area with a key focus on measurement of air-quality in terms of PM2.5. Indeed, the studies on this scope are important and tend to indicate some remerkable outputs. However, the current paper has a great limitation in terms of novelty anf scope. This work installes some sensors and collect information from these sensors, resulting in a city map. I respect this work a lot, but it fails to be published in a high-prestigous journal. 

It is expected from a typical good journal-paper that a new method should be designed or a popular method is applied with significant improvements. Also, performance comparisons with well-known approaches are necessary to assess the effevtiveness of the work. 

Author Response

Review Report 4

Comments and Suggestions for Authors

This paper yields an interesting area with a key focus on measurement of air-quality in terms of PM2.5. Indeed, the studies on this scope are important and tend to indicate some remerkable outputs. However, the current paper has a great limitation in terms of novelty anf scope. This work installes some sensors and collect information from these sensors, resulting in a city map. I respect this work a lot, but it fails to be published in a high-prestigous journal. 

It is expected from a typical good journal-paper that a new method should be designed or a popular method is applied with significant improvements. Also, performance comparisons with well-known approaches are necessary to assess the effevtiveness of the work. 

Authors’ answer: We are surprised by the reviewer comments, which are on the opposite of those from the other reviewers. Firstly, performances comparison have been always conducted in Paris, and new comparison with  UBA reference station are presented here. Secondly, it is totally a new method of measurements (sensors permanently on buses for PM2.5 detection for at least one year). PM2.5 maps with such spatial resolution have never been analyzed and published before. Thirdly, we discussed the interest of such new method of measurements to better evaluate the real expose of the citizens to local pollution.

Round 2

Reviewer 1 Report

Comments and Suggestions for Authors

It would be helpful to include more studies

Author Response

Reviewer 2:

It would be helpful to include more studies

Authors answer: We understand the reviewer comments. As said previously, it seems difficult to go further with the results already available. It is why we speak of the future analysis with new results in the conclusion.

Reviewer 4 Report

Comments and Suggestions for Authors

I am surprised by the author comments for their misunderstanding technically. Performance comparisons are conducted with at least one state-of-the-art approach, not with a city parameter. Composing a system by mounting a set of sensors is not a new method, it is an experimental setup. Nevertheless, the scope of experimental setup and the deep analyse in this paper can be considered as a sufficient novelty. However, you need to provide more details about prediction model used along with its math/analytical model. For a novel work, you can improve the prediction model based on your system dynamics. 

Author Response

Reviewer 4:

I am surprised by the author comments for their misunderstanding technically. Performance comparisons are conducted with at least one state-of-the-art approach, not with a city parameter. Composing a system by mounting a set of sensors is not a new method, it is an experimental setup. Nevertheless, the scope of experimental setup and the deep analyse in this paper can be considered as a sufficient novelty. However, you need to provide more details about prediction model used along with its math/analytical model. For a novel work, you can improve the prediction model based on your system dynamics. 

Authors answer: We agree with the reviewer that we have used a new experimental set up. It seems reasonable that the readers want to know how the measurements were locally validated for this case-study, although more validation work was done previously. We have added in the text: “The federal office for environmental protection of Germany (Umweltbundesamt, UBA) relies on measuring stations with established and technology validated for regulatory purposes. Data from these stations can therefore be used for validation of other sensors.”

Our aim was not to work on a new modeling approach, but to present the consistency for the same location between Pollutrack sensors, a reference instrument and modeling data. But we agree that our results should help to further improve the modeling approaches. We have also added in the text and the corresponding reference: “This approach originally focused on PM10 measurements, and was subsequently adapted to the study of PM2.5. It has been validated in [22] by comparing results with 11 models’ outputs published between 2000 and 2021.”